# Diagnosis and Prediction of Endometrial Carcinoma Using Machine Learning and Artificial Neural Networks Based on Public Databases

**DOI:** 10.3390/genes13060935

**Published:** 2022-05-24

**Authors:** Dongli Zhao, Zhe Zhang, Zhonghuang Wang, Zhenglin Du, Meng Wu, Tingting Zhang, Jialu Zhou, Wenming Zhao, Yuanguang Meng

**Affiliations:** 1Department of Obstetrics & Gynecology, Chinese People’s Liberation Army (PLA) Medical School, No. 28, Fuxing Road, Haidian District, Beijing 100853, China; zz18813066251@163.com (D.Z.); 17865190928@163.com (T.Z.); kalozzhou@163.com (J.Z.); 2Department of Obstetrics and Gynecology, Seventh Medical Center of Chinese PLA General Hospital, No. 5, Nanmencang, Dongsishitiao, Dongcheng District, Beijing 100700, China; zhangzhe301@126.com; 3National Genomics Data Center & CAS Key Laboratory of Genome Sciences and Information, Beijing Institute of Genomics, Chinese Academy of Sciences and China National Center for Bioinformation, Building 104, Courtyard 1, Beichen West Road, Chaoyang District, Beijing 100101, China; wangzhonghuang17m@big.ac.cn (Z.W.); duzhl@big.ac.cn (Z.D.); 4University of Chinese Academy of Sciences, 19 Yuquan Road (a), Shijingshan District, Beijing 100049, China; 5Medical College, Graduate School of Nankai University, No. 94, Weijin Road, Nankai District, Tianjin 300110, China; awumeng_1992@163.com; 6Department of Gynecology and Obstetrics, Chinese PLA General Hospital, No. 28, Fuxing Road, Haidian District, Beijing 100853, China

**Keywords:** endometrial carcinoma, GEO, TCGA, random forest, receiver operating characteristic curve

## Abstract

Endometrial carcinoma (EC), a common female reproductive system malignant tumor, affects thousands of people with high morbidity and mortality worldwide. This study was aimed at developing a prediction model for the diagnosis of EC in the general population. First, we obtained datasets GSE63678, GSE106191, and GSE115810 from the Gene Expression Omnibus (GEO) database, dataset GSE17025 from the GEO database, and the RNA sequence of EC from The Cancer Genome Atlas (TCGA) database to constitute the training, test, and validation groups, respectively. Subsequently, the 96 most significantly differentially expressed genes (DEGs) were identified and analyzed for function and pathway enrichment in the training group. Next, we acquired the disease-specific genes by random forest and established an artificial neural network for the diagnosis. Receiver operating characteristic (ROC) curves were utilized to identify the signature across the three groups. Finally, immune infiltration was analyzed to reveal tumor-immune microenvironment (TIME) alterations in EC. The top 96 DEGs (77 down-regulated and 19 up-regulated genes) were primarily enriched in the interleukin-17 signaling pathway, protein digestion and absorption, and transcriptional misregulation in cancer. Subsequently, 14 characterizing genes of EC were identified by random forest. In the training, test, and validation groups, the artificial neural network was constructed with high diagnostic accuracies of 0.882, 0.864, and 0.839, respectively, and areas under the ROC curve (AUCs) of 0.928, 0.921, and 0.782, respectively. Finally, resting and activated mast cells were found to have increased in TIME. We constructed an artificial diagnostic model with excellent reliability for EC and uncovered variations in the immunological ecosystem of EC through integrated bioinformatics approaches, which might be potential diagnostic targets for EC.

## 1. Introduction

Endometrial carcinoma (EC), a malignancy of the inner epithelial lining of the uterus, is a common neoplasm in women worldwide, with increasing rates of incidence and disease-associated mortality in recent years [1,2], seriously threatening women’s physical and mental health. Most cases of early EC are cured by surgery alone or with adjuvant therapy. However, many cases of EC are diagnosed in the advanced stage at the first consultation and are associated with a poor prognosis. Although the survival rate of patients has increased, owing to molecular targeted therapy, no targeted gene mutations have been explored in advanced EC [3,4,5].

Currently, EC is diagnosed mainly based on clinical symptoms; physical findings; results of laboratory investigations, transvaginal ultrasound, pelvic ultrasonography, endometrial biopsy with hysteroscopy, and imaging (computed tomography, positron emission tomography/computed tomography, and magnetic resonance imaging); and some biomarkers (e.g., CA125 and HE4) [6,7,8,9]. The purpose of these investigations is to examine the endometrial cells, determine the disease extent, and detect the presence/absence of metastasis. Although these methods have good sensitivity for the diagnosis of EC, they have disadvantages, such as poor specificity (particularly transvaginal ultrasound), invasiveness, pain, and high cost. Therefore, improved examination techniques are urgently required, and target genes seem to be appropriate candidates.

Owing to advancements in computer technology and the introduction of sequencing technology, studies have promoted our understanding of cellular and genetic changes during oncogenesis and yielded more targeted and individualized treatment choices [10,11,12]. Machine learning, a component of artificial intelligence, using computer technology to simulate human intellect, can make predictions using mathematical algorithms after being trained with data. Deep learning, a branch of machine learning, focuses on making forecasts using a multilayer neural network algorithm and can expand model predictions exponentially with increased data volume and dimension, making it suitable for large-scale data analyses. Thus, deep learning can generate meaningful insights and discern relevant traits from genomic data. Genomic analyses have revealed novel biological targets for EC. The genetic bases of cancer progression and therapeutic response have been extensively studied, and the developments of next-generation sequencing and machine learning have yielded opportunities to systematically assess differentially expressed genes (DEGs) [11,12,13]. Moreover, large public databases, such as the Gene Expression Omnibus (GEO) and The Cancer Genome Atlas (TCGA), have provided abundant cancer genome sequencing data, which have improved our understanding of molecular changes in oncogenesis. However, due to the lack of multi-omics data, studies on the genomic analysis of EC focusing on gene expression or immune response are few. Since RNA sequencing of tumor tissues is usually performed to characterize gene expression and tumor immune microenvironment (TIME) cells, many datasets have estimated the abundance of DEGs and TIME cells in neoplastic tissue [13,14,15,16].

This study was aimed at identifying the signature genes in EC using machine learning, constructing a diagnostic model using an artificial neural network, and verifying the model in three EC cohorts. Finally, the changes in TIME during EC were confirmed.

## 2. Materials and Methods

### 2.1. Data Collection and Pre-Processing

Table 1 demonstrates the datasets utilized in this study. The gene expression datasets GSE63678, GSE106191, and GSE115810 were obtained from GEO (https://www.ncbi.nlm.nih.gov/geo/) (accessed on 3 March 2022), merged, and corrected for the batch effect to constitute the training group. The dataset GSE17025 from GEO and the gene expression of EC from TCGA (https://www.cancer.gov/) (accessed on 3 March 2022) were accessed to constitute the test and validation groups, respectively. Our study complied with the publication guidelines laid down by GEO and TCGA. No ethics committee approval was required.

### 2.2. Exploration of DEGs and Functional Enrichment

The 96 DEGs across the EC and para-cancer samples in the training group were calculated using the R package “limma”, which employed the empirical Bayesian method and the moderated Wilcox test to assess differences in gene expression. Subsequently, heatmaps and volcanic maps were drawn using the R package with an absolute log2 fold change ≥0.8 and an adjusted *p*-value < 0.05. For the next functional analysis of the 96 DEGs in EC, the R package “clusterProfiler” was used to perform the Gene Ontology (GO) and Kyoto Encyclopedia of Genes and Genomes (KEGGs) enrichment analyses. The GO analysis mainly comprised the biological process, cellular component, and molecular function. For the functional enrichment analysis, statistical significance was set at *p* < 0.05, and the R packages “enrichplot” and “ggplot2” were used.

### 2.3. Construction of Metascape and the Protein-Protein Interaction (PPI) Network

In addition, we also analyzed gene sets using the online toolkit WebGestalt (http://www.webgestalt.org/) (accessed on 3 March 2022); performed enrichment analyses using Metascape (http://metascape.org/) (accessed on 3 March 2022), Reactome, and WikipathwayCancer; and investigated a protein–protein interaction (PPI) network using the STRING (https://cn.string-db.org/) (accessed on 3 March 2022) database.

### 2.4. Selection of the Signature Genes and Construction of the Diagnostic Prediction Model

Random forest analyses were performed, and characteristic DEGs were selected based on the point at which the error of cross validation was the least. The setting seed was 123,456, and the ntree was 500. Subsequently, the characteristic genes were assigned a gene importance score, and those with a score >0.9 were selected and visualized by the R packages “limma” and “pheatmap”. Next, we clustered the samples according to the expression of DEGs in the training group and found that the samples were divided into two clusters, similar to carcinoma and paraneoplastic samples.

Subsequently, we assigned scores to the specific DEGs to eliminate batch effects in samples. Up-regulated genes greater than the median value were scored 1, whereas the rest were scored 0; similarly, down-regulated genes lesser than the median value were scored 1, whereas the rest were scored 0. The artificial neural network model for the EC diagnosis was constructed from three types of layers: the input layer, with the scores of 14 genes; the hidden layers, with the scores and weights of genes; and the output layer, with the results for control and experimental samples. The R package “NeuralNetTools” was applied for the procedure with a seed of 12,345,678. Similarly, the selected DEGs and the constructed artificial neural network were applied to the test and validation groups. Unlike the other two groups, the control samples enrolled in the test group comprised tissues of other uterine pathologic types, whereas the samples in the experimental group comprised tissues of early EC. In addition, we constructed a receiver operating characteristic (ROC) curve using the R package “pROC” and assessed the area under the ROC curve (AUC) for the diagnostic model across the three cohorts.

### 2.5. Identification of TIME

In the analysis of immune cell infiltration, a total of 22 immune cells were identified by the CIBERSORT algorithm and screened using the R packages “e1071”, “preprocessCore”, and “CIBERSORT.R” at *p* < 0.05. The correlation between the immune cells was calculated using the R package “corrplot.” Moreover, the different distribution of immune cells between EC and normal tissues was measured and presented as a violin plot.

## 3. Results

### 3.1. DEGs and Functional Enrichment Analysis Results in EC

Based on the filer criteria, a total of 96 DEGs were found between EC and normal samples in the training group and analyzed. There were 19 up-regulated (e.g., MMP12 and CCL20) and 77 down-regulated (e.g., SFP4, OGN, OSR2, FOXL2, and IGFBP4) genes (Figure 1A,B). The top 10 GO terms revealed that the DEGs were mainly involved in collagen-containing extracellular matrix organization and signaling receptor activator activity (Figure 1C). KEGG terms demonstrated that the 96 DEGs were mainly involved in the interleukin-17 (IL-17) signaling pathway, protein digestion and absorption, and transcriptional misregulation in cancer (Figure 1D); thereby playing important roles in inflammatory and immune processes and the occurrence and development of tumors. All enrichment analysis results were closely related to TIME.

### 3.2. Metascape and PPI Network Analysis Results

A network diagram was created based on Metascape analysis. Spots represented functions or pathways. Larger and connected points represented the presence of more similar genes between the functions or pathways. The NABA_CORE_MATRISOME gene set contained many genes encoding extracellular matrix organization and extracellular matrix-associated proteins activated in EC, while the NABA_MATRISOME_ASSPCIATED gene set contained many genes encoding vascular development, tissue morphogenesis, and growth regulation (Figure 2A). Figure 2B shows the top 50 function enrichments. Subsequently, the enrichment analyses of DisGeNET and PaGenBase revealed that the DEGs were primarily specialized in endometrial neoplasms and the uterus (Figure 2C,D), consistent with this study. During the pathogenesis of EC, epigenetic changes in pathogenic genes were mainly regulated by transcription factors EP300, RELA, JUN, SP1, NFKB1, ERG, HDAC1, CEBPA, FOS, and HIF1A (Figure 2E), which play important roles in inflammation, cell proliferation, transformation, differentiation, apoptosis, and immune response. In addition, the PPI network showed a relationship between different genes and proteins in the three sub-modules (Figure 2F). The NABA_CORE_MATRISOME sub-module included COL21A1, COL5A1, COL6A2, COL3A1, and COL15A1, which can identify the structural components of the extracellular matrix to provide tensile strength; the extracellular matrix organization sub-module included SPP1, IGFBP4, GAS6, MXRA8, and SPARCL1, which could enable proteins and/or the extracellular matrix; the NABA_MATRISOME_ASSOCIATED sub-module included P2RY14, CXCL8, CCL20, CXCL3, and CXCL12, which could enable protein binding and chemokine activity.

### 3.3. Exploration of Characteristic DEGs and Diagnostic Prediction Model of EC

We conducted a random forest analysis to identify the characteristic DEGs. The black line and horizontal and vertical axes represented the error value of the samples, number of trees, and cross-validation error, respectively (Figure 3A). Figure 3B shows the importance of genes. After re-validating DEGs, all 14 EC-signature DEGs with a score >0.9 were enrolled, including three up-regulated (MMP12, MMP9, and ADAMDEC1) and 11 down-regulated (OGN, FOXL2, IGFBP4, DCHS1, ENPP2, ALDH1A2, ADAMTS5, MXRA8, EFEMP1, EFS, and ENPEP genes (Figure 3C and Figure 4). In the diagnostic prediction model, the control and experimental samples were aggregated, which signified that the expression of the pathogenic genes was distinguished between the normal and EC samples (Figure 3D). In addition, for the training, test, and validation groups, the AUCs were 0.928, 0921, and 0.782, respectively, and the accuracies were 0.882, 0.864 and 0.839, respectively (Figure 5A–C, Table 2); implying that the EC diagnostic prediction model could be used as an independent diagnostic predictor of EC.

### 3.4. TIME of EC

Figure 6A shows the 22 categories of immunocytes in each sample. Resting and activated mast cells, neutrophils, macrophage M1s, activated NK cells, and eosinophils were relatively abundant in EC. Figure 6B shows the correlation in infiltration of immune cells. The greater the absolute value of the number, the stronger the correlation coefficients, with red and blue colors representing positive and negative correlations, respectively. Activated and resting mast cells showed a strong negative correlation, with a correlation coefficient of −0.54. Activated mast cells and NK cells showed a negative correlation, with a correlation coefficient of −0.43. The activated T cells CD4 and CD8 showed a strong positive correlation, with a correlation coefficient of 0.36 (Figure 6B). In summary, resting and activated mast cells, neutrophils, macrophage M1s, activated NK cells, and eosinophils in EC and normal samples were significantly different (Figure 6A–C); high expressions of activated mast cells, macrophage M1, and neutrophils and low expressions of resting mast cells, activated NK cells, and eosinophils were found in EC.

## 4. Discussion

At present, EC is diagnosed mainly based on clinical symptoms, physical findings, results of laboratory investigations, and imaging examination. Endometrial biopsy under hysteroscopy seems to be the best method for the diagnosis of benign EC [17,18]. Fertility retention technology can effectively improve the quality of life of gynecological cancer patients, and has become the goal and hope for cancer survivors to live a better life [19]. Studies based on systems biology proteomics have highlighted the exact potential molecular mechanisms associated with SLN and EC grades [20,21]. The aim of these investigations is to examine the endometrial cells, determine the disease extent, and detect the presence/absence of metastasis. Although the accuracy of the diagnosis and treatment of EC has made great progress in recent years, the molecular mechanism remains unknown. Abnormal gene expression and immune response in TIME play active roles in tumor occurrence, development, invasion, metastasis, and recurrence and are key considerations influencing tumor prognosis [16,22,23,24]. Endometrial biopsy under hysteroscopy seems to be the best method for the diagnosis of benign EC. In this study, we focused on transcriptional data from GEO and TCGA to identify the complex correlations of the signature genes for EC with a diagnosis and to build a diagnostic prediction model of EC, involving 14 signature genes by random forest and artificial neural network analyses, which distinguished patients with EC from the general population to guide diagnosis and treatment.

We obtained 96 DEGs, including 19 up-regulated and 77 down-regulated genes, and investigated sophisticated biological functions using GO and KEGG analyses in the training group. The outcome indicated that the DEGs were mainly enriched in extracellular matrix and structure organizations, and involved in the IL-17 signaling pathway, protein digestion and absorption, and transcriptional mis-regulation in TIME. These results indicated that changes in gene expression could be conducive to tumor remodeling and promote chronic inflammation, tumor progression, metastasis, and immune escape. We also obtained ten transcription factors, including EP300, RELA, JUN, SP1, NFKB1, ERG, HDAC1, CEBPA, FOS, and HIF1A, which regulated gene expression and played important roles in inflammation, cell proliferation, transformation, differentiation, apoptosis, and immune response [25,26,27,28,29,30]. In addition, the PPI network mainly showed a relationship between different genes and proteins among the three sub-modules. The NABA_CORE_MATRISOME sub-module comprised COL21A1, COL5A1, COL6A2, COL3A1, and COL15A1, which could identify structural components of the extracellular matrix to provide tensile strength. The extracellular matrix organization sub-module comprised SPP1, IGFBP4, GAS6, MXRA8, and SPARCL1, which could enable proteins and extracellular matrix. The NABA_MATRISOME_ASSOCIATED sub-module comprised P2RY14, CXCL8, CCL20, CXCL3, and CXCL12, which could enable protein binding and chemokine activity.

To obtain a good neural network model, we found 14 characteristic genes for EC by the machine learning method random forest. A diagnostic prediction model for EC was constructed using the artificial neural network, which may be widely applied to the formulation of diagnosis and treatment models for EC. In the model, expressions of MMP12, MMP9, and ADAMDEC1 were increased in EC, and those of OGN, FOXL2, IGFBP4, DCHS1, ENPP2, ALDH1A2, ADAMTS5, MXRA8, EFEMP1, EFS, and ENPEP were decreased in EC. MMP12 and MMP9 were related to cancer development, progression, and survival through various pathological processes and play essential roles in tumor invasion and metastasis [31,32,33,34]. Therefore, MMP12 knockdown inhibited proliferation and invasion of nasopharyngeal and lung cancers. Overexpression of ADAMDEC1 is correlated with tumor progression, inflammation, immunotherapeutic response, and a poor prognosis in many cancers [35,36,37]. Under-expressed OGN and EFS, compared to the normal samples, improved survival, reduced tumor recurrence, and reversed the epithelial to mesenchymal transition by inhibiting EGFR/AKT/Zeb-1 in tumors [38,39]. In a previous study, FOXL2 was considered for molecular diagnostic testing in ovarian adult granulosa cell and microcystic stromal cancers [40]. IGFBP-4 plays an important role in tumor growth regulation by inhibiting IGF actions [41]. Although these feature genes are widely expressed in tumors, according to previous reports, further research is required to clarify the gene function in the pathology of carcinoma, particularly EC. According to the traditional model, EC is divided into types 1 and 2, with certain classic mutations between the two types. Type 1 has mutations in PTEN, ARID1A, PIK3CA, and KRAS, while type 2 has mutations in TP53. Currently, EC is mainly diagnosed based on uterine curettage or biopsy findings. Some data suggest that the susceptibility of endometrial biopsy for EC is 52–94% [42,43,44,45,46]. The accuracy of differentiation of EC in other studies was slightly lower than our model (Table 2) [21,47,48]. Particularly, the test group comprised non-cancerous uterine pathologic types and early EC. The diagnostic rate of 100% in the non-cancerous group confirmed the efficacy of our diagnostic model for early EC in the test group. Thus, the model in the training and test groups showed a good effect, while that in the validation group showed an average effect. The 14 feature genes were key potential biomarkers of EC, but further studies are required to verify the results.

In addition, we also focused on TIME of EC and found that high expressions of activated mast cells, macrophage M1s, and neutrophils, and low expressions of resting mast cells, NK cells, and activated eosinophils played vital roles in EC. Multiple studies have documented that mast cells, neutrophils, macrophage M1s, NK cells, and eosinophils play a protective role during cancer progression, such as inflammatory responses, development of blood vessels, apoptosis, proliferation, invasion, and immune evasion [49,50,51,52,53,54,55,56].

However, this study has some limitations. First, the RNA sequencing data were only obtained from public databases. Second, although we validated the predictive performance of the EC diagnosis, further investigation is required for accurate validation. Further basic and clinical studies should be performed to validate the outcome and find a simpler, faster, and more economic approach.

## 5. Conclusions

In our study, we identified 14 genes involved in EC, verified them, based on GEO and TCGA, and established a robust diagnostic prediction model for EC through an artificial neural network, which was promising for the exploration of new diagnostic tools. The diagnostic model possessed excellent sensitivity and specificity, demonstrating the capability of diagnosing early EC. We also discovered that activated and resting mast cells were important and inversely correlated in EC. These results could serve as a basis for extensive cohorts in the future.

## Figures and Tables

**Figure 1 genes-13-00935-f001:**
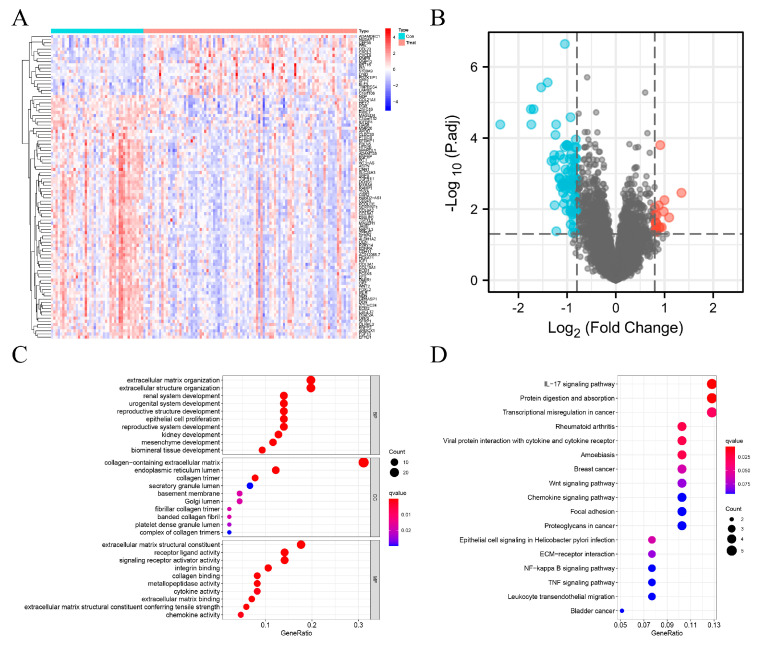
Identification of 96 DEGs in EC in the training group. (**A**) Heatmap of DEGs. The columns represent samples, and rows represent genes. The red color represents up-regulation, and the blue color represents down-regulation. |log2FC| > 0.8, *p*-value < 0.05. (**B**) Volcanic map of DEGs. The red, blue, and black colors represent up-regulated, down-regulated, and undifferentiated genes, respectively. |log2FC| > 0.8, *p*-value < 0.05. (**C**) Top 10 biological processes, cellular components, and molecular functions with the most significant *p*-value. (**D**) All KEGG enrichment results of DEGs.

**Figure 2 genes-13-00935-f002:**
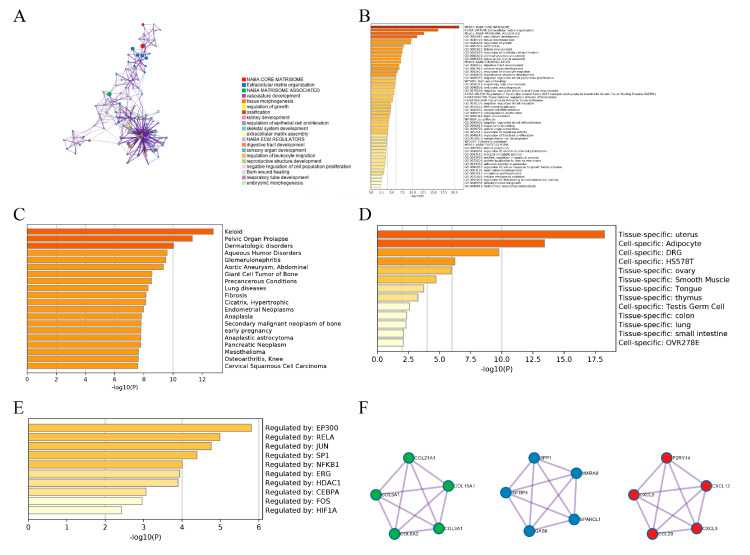
PPI network based on Metascape. (**A**) Network diagrams of the enrichment pathway and process of EC. (**B**) Bar plot of the enrichment pathway and process of EC. (**C**) Bar plot of enrichment on DisGeNET. (**D**) Bar plot of enrichment on PaGenBase. (**E**) Bar chart of enrichment on TRRUST. (**F**) Three sub-modules of PPI.

**Figure 3 genes-13-00935-f003:**
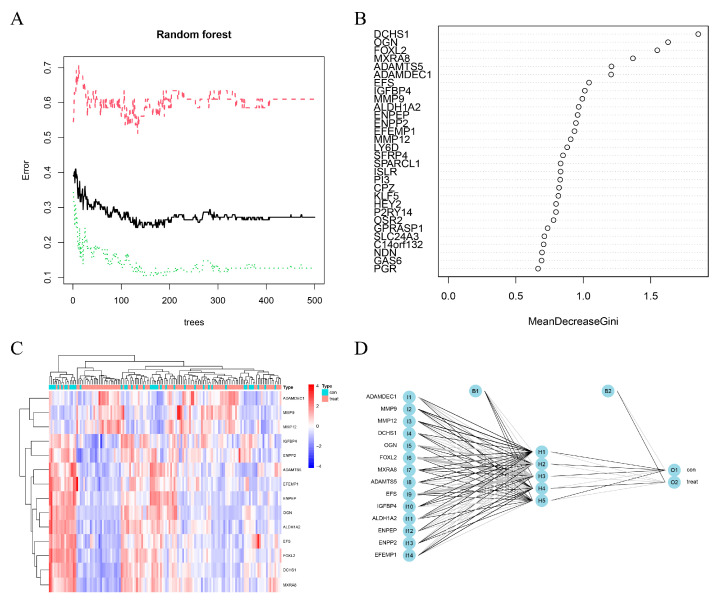
Selection of signature genes by machine learning and construction of a diagnostic prediction model by artificial neural network. (**A**) Construction of random forest. (**B**) Exploring signature genes of EC based on gene importance scores. (**C**) Heatmap of 14 characteristic DEGs. (**D**) Process of constructing artificial neural network.

**Figure 4 genes-13-00935-f004:**
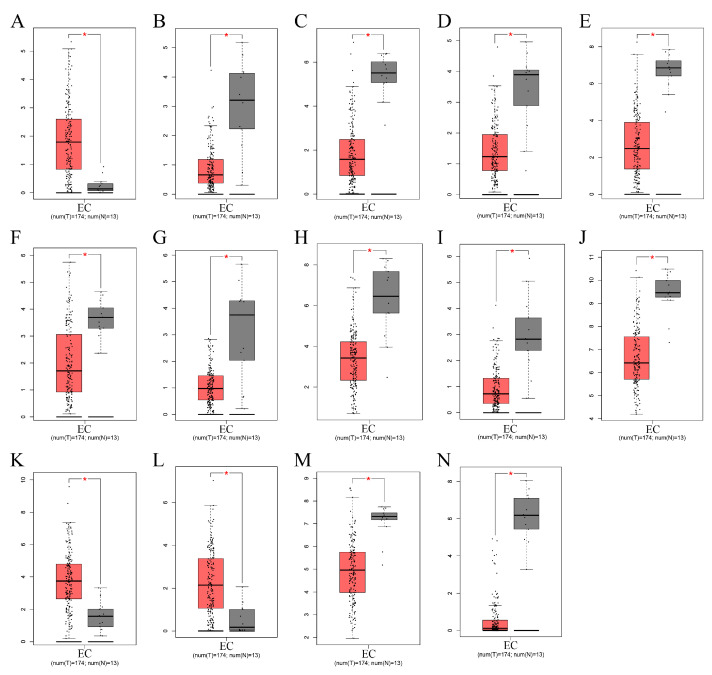
Box diagram of 14 characteristic genes in EC and healthy controls with *p*-value < 0.01. (**A**–**N**) ADAMDEC1, ADAMTS5, ALDH1A2, DCHS1, EFEMP1, EFS, ENPEP, ENPP2, FOXL2, IGFBP4, MMP9, MMP12, MXRA8, and OGN. The red color represents EC, and the black color represents healthy controls. * means *p*-value < 0.01.

**Figure 5 genes-13-00935-f005:**
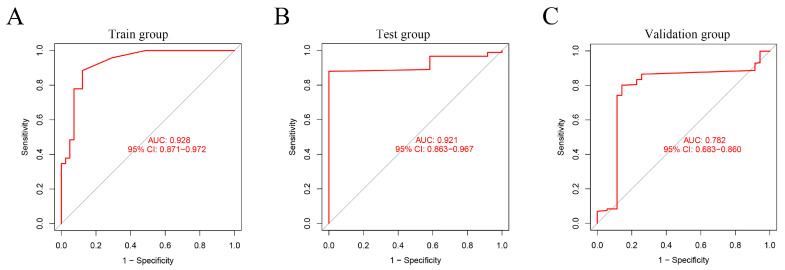
ROC curves of the three groups. (**A**) Training group. (**B**) Test group. (**C**) Validation group.

**Figure 6 genes-13-00935-f006:**
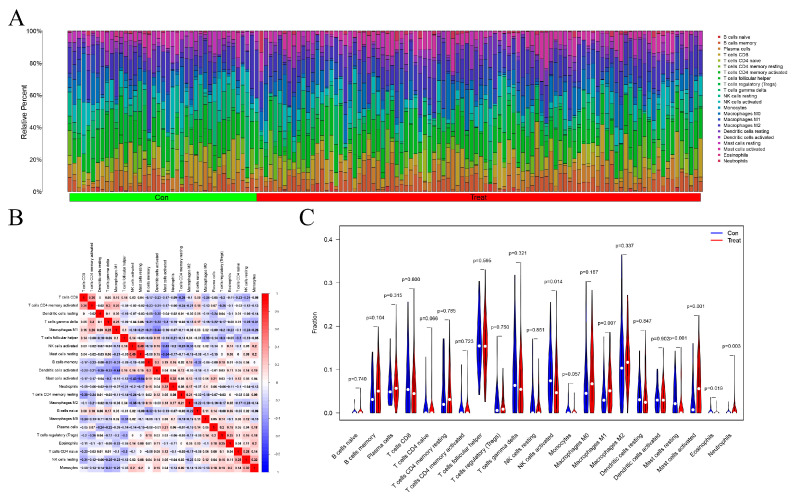
Tumor-immune microenvironment of EC. (**A**) Histogram of 22 types of immune cells in EC and healthy controls. (**B**) Correlation of immune cells in EC. (**C**) Violin image of immune cells.

**Table 1 genes-13-00935-t001:** Composition of the datasets and component of patients enrolled in this study.

	Train Group			Test Group	Validation Group
	GSE106191	GSE115810	GSE63678	GSE17025	TCGA
Sample Count	97	27	35	103	583
Normal	64	3	5	12	35
Cancer	33	24	7	91	548
Enrollment	97	27	12	103	583

**Table 2 genes-13-00935-t002:** Neural Diagnostic for the training, test and validation cohorts.

		Training Group		Test Group		Validation Group	
		Normal	Cancer	Normal	Cancer	Normal	Cancer
Prediction results	Normal	32	7	12	14	26	85
	Cancer	9	88	0	77	9	463
	Normal Accuracy	0.780		1.000		0.743	
	Cancer Accuracy	0.926		0.846		0.845	
	Accuracy	0.882		0.864		0.839	

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
