# Peer review of "Diagnosis and Prediction of Endometrial Carcinoma Using Machine Learning and Artificial Neural Networks Based on Public Databases"

_genes, 2022, doi:10.3390/genes13060935_

Round 1
Reviewer 1 Report
I read with great interest the Manuscript titled “Diagnosis and Prediction of Endometrial Carcinoma Using Machine Learning and Artificial Neural Networks Based on Public Databases", which falls within the aim of Genes.
In my honest opinion, the topic is interesting enough to attract the readers’ attention. Methodology is accurate and conclusions are supported by the data analysis. Nevertheless, authors should clarify some points and improve the discussion citing relevant and novel key articles about the topic.
Authors should consider the following recommendations:
- Manuscript should be further revised by a native English speaker
- Please incorporate tables and figures in the main text.
- Improve discussion by comparison with existing literature about the following themes:
- - Fertility sparing approach in endometrial cancer (see PMID:32419847)
- - Hysteroscopy in the management of endometrial cancer (PMID: 35117379 and 35413062)
- - Non-invasive Diagnostic approaches in endometrial cancer (PMID: 34218728 and 35385178)
- - ML and proteomics of sentinel lymphnode in endometrial cancer (PMID 35117375 and 34195683)
Author Response
We greatly appreciate the input, and we've enclosed a revised version that will hopefully address the concerns in the reviews, to which we detail our responses below.
Point 1: Manuscript should be further revised by a native English speaker
Response 1: Please provide your response for Point 1. (in red)
Reply: Thank you very much for your valuable opinions. We sent the manuscript to a professional editing company for English editing to improve the article for language and style. The edit was performed by professional editors at MJEditor (www.mjeditor.com), which also provided a language editing certificate. In addition, during the revision process, we have consulted a number of teachers with higher English proficiency to make the manuscript better and suitable for publication.
Point 2: Please incorporate tables and figures in the main text.
Response 2: Please provide your response for Point 2. (in red)
Reply: Thank you for your professional advices. We have incorporated tables and figures in the main text. The specific modifications have been marked in red letters in the manuscript.
Point 3: Improve discussion by comparison with existing literature about the following themes:
- Fertility sparing approach in endometrial cancer (see PMID:32419847)
- Hysteroscopy in the management of endometrial cancer (PMID: 35117379 and 35413062)
- Non-invasive Diagnostic approaches in endometrial cancer (PMID: 34218728 and 35385178)
- ML and proteomics of sentinel lymphnode in endometrial cancer (PMID 35117375 and 34195683)
Response 3: Please provide your response for Point 3. (in red)
Reply: Thank you very much for your approval of our manuscript and for your professional advice. We are sorry to have missed such high-quality references. Based on the literature you provided, we have revised the relevant discussion as follows:
At present, EC is diagnosed mainly based on clinical symptoms; physical findings; results of laboratory investigations, imaging examination. Endometrial biopsy under hysteroscopy seems to be the best method for the diagnosis of benign EC[17, 18]. Fertility retention technology can effectively improve the quality of life of gynecological cancer patients, and become the goal and hope for cancer survivors to live a better life[19]. Studies based on systems biology proteomics have highlighted the exact potential molecular mechanisms associated with SLN and EC grades[20, 21]. The aim of these investigations is to examine the endometrial cells, determine the disease extent, and detect the presence/absence of metastasis.

Reviewer 2 Report
The authors have a great internal team to support the advancement of cutting-edge science. The purpose of this study is to identify the signature genes in endometrial cancer using machine learning, constructing a diagnostic model using an artificial neural network, and verifying the model in three different cohorts. I can see their strong communication and analytical skills, as well as motivation to empower a team and to work independently.
Author Response
First, many thanks to Referee, we're glad you appreciated the manuscript, and many thanks for your time and input. In response to your specific comments.
Point 1: The authors have a great internal team to support the advancement of cutting-edge science. The purpose of this study is to identify the signature genes in endometrial cancer using machine learning, constructing a diagnostic model using an artificial neural network, and verifying the model in three different cohorts. I can see their strong communication and analytical skills, as well as motivation to empower a team and to work independently.
Response 1: Please provide your response for Point 1. (in red)
Thank you very much for your approval of our manuscript and for your valuable opinions.

Round 2
Reviewer 1 Report
I am pleased to review the revised version of the enclosed manuscript. The authors have solved all the issues highlighted in the previous version. I have no further concerns.